# Bridging Radiotherapy to Immunotherapy: The IFN–JAK–STAT Axis

**DOI:** 10.3390/ijms222212295

**Published:** 2021-11-14

**Authors:** Lewis Zhichang Shi, James A. Bonner

**Affiliations:** 1Department of Radiation Oncology, University of Alabama at Birmingham, Birmingham, AL 35233, USA; 2Department of Microbiology, University of Alabama at Birmingham, Birmingham, AL 35233, USA; 3Programs in Immunology, University of Alabama at Birmingham, Birmingham, AL 35233, USA; 4O’Neal Comprehensive Cancer Center, University of Alabama at Birmingham, Birmingham, AL 35233, USA

**Keywords:** radiotherapy, immunotherapy, interferon, IFN-γ, JAK, STAT

## Abstract

The unprecedented successes of immunotherapies (IOs) including immune checkpoint blockers (ICBs) and adoptive T-cell therapy (ACT) in patients with late-stage cancer provide proof-of-principle evidence that harnessing the immune system, in particular T cells, can be an effective approach to eradicate cancer. This instills strong interests in understanding the immunomodulatory effects of radiotherapy (RT), an area that was actually investigated more than a century ago but had been largely ignored for many decades. With the “newly” discovered immunogenic responses from RT, numerous endeavors have been undertaken to combine RT with IOs, in order to bolster anti-tumor immunity. However, the underlying mechanisms are not well defined, which is a subject of much investigation. We therefore conducted a systematic literature search on the molecular underpinnings of RT-induced immunomodulation and IOs, which identified the IFN–JAK–STAT pathway as a major regulator. Our further analysis of relevant studies revealed that the signaling strength and duration of this pathway in response to RT and IOs may determine eventual immunological outcomes. We propose that strategic targeting of this axis can boost the immunostimulatory effects of RT and radiosensitizing effects of IOs, thereby promoting the efficacy of combination therapy of RT and IOs.

## 1. Introduction

IOs have revolutionized the field of cancer care over the past decade [1]. By enhancing anti-tumor immunity, IOs induce impressive long-term clinical benefits in patients with various forms of late-stage cancers [2,3,4,5,6], many of which were not amenable to any treatment not long ago. This also revives strong interests in understanding the immunological effects of well-established therapeutic modalities such as radiotherapy (RT). RT has been a mainstay cancer therapy since the 1950s, with more than 50% of patients receiving this treatment during their course of illness. Historically, RT has been included in immunosuppressive conditioning of patients for hematopoietic stem cell transplantation, given its myeloablative action and the radiosensitivity of quiescent lymphocytes. However, recent studies have provided compelling evidence that RT also induces immunostimulatory effects. As such, the 6th R in RT was recently coined as the Reactivation of the immune system [7], adding to the other 5Rs known to be important in determining the final clinical outcomes of RT: Repair, Re-assortment, Repopulation, Reoxygenation, and Radiosensitivity. Ironically, despite this recent fervent interest in RT-induced immunomodulation, a close link between RT and the immune system was recognized a long time ago by Russ and Murphy [8], but it had been largely neglected [9]. By inducing immunogenic cell death (ICD), RT leads to the release of damaged DNA and damage-associated molecular patterns (DAMPs) such as calreticulin (CRT), ATP, and high mobility group box protein 1 (HMGB1) [10,11], which can act as immunogenic adjuvants. In addition, RT (especially, hypofractionated RT) induces “novel” mutations/neoantigens, promoting the antigenicity of tumors. Together, these signals drive maturation and upregulation of MHC class I and II molecules on dendritic cells (DCs), the professional antigen presenting cells [12], which in turn prime tumor-reactive T cells and recruit them into the tumor microenvironment (TME) to combat tumor cells [13].

With these immunogenic effects and technical advances in RT such as three-dimensional image guidance, modulation of beam intensity, and multileaf collimators, precision RT (e.g., stereotactic ablative radiotherapy (SABR) and intensity-modulated radiation therapy (IMRT)) has been developed, which allows conformal delivery of larger doses per fraction while minimizing inadvertent doses to normal tissues. This makes RT an appealing therapeutic option, particularly in this era of personalized medicine and IOs, as they can turn tumors into in situ “personalized” vaccines. Moreover, by directly killing tumor cells, RT effectively debulks large tumors that are typically not responsive to IOs and potentially resensitizes them to IOs. These enticing properties endow RT as an “ideal” therapeutic partner for IOs. In support of this concept, clinical trials have shown improved efficacy from combination therapy of RT and IOs [14,15,16]. While these results have been promising, overall efficacy from these clinical trials is still limited. Therefore, many studies have been conducted to delineate the molecular mechanisms underscoring the immunomodulatory effects of RT and IOs, in order to better employ these treatments in the future.

Generally speaking, RT and IOs rely on their abilities to kill or suppress tumor cells, thus preventing tumor progression. For this reason, it is not surprising that our systematic literature search revealed that evolutionally conserved molecular pathways essential for cell division and differentiation are required for RT and IOs. These include hormone-driven nuclear receptor pathways (e.g., androgen and estrogen receptor pathways) [17], developmental pathways (e.g., the Wnt, Hedgehog, TGF-β, and Notch) [18], the inflammatory NF-κB pathway [19], the growth factor-regulated signaling pathways (e.g., PI3K-AKT-mTOR and MAPK) [20], and the prevalent JAK (Janus Kinase)-STAT (Signal Transducer and Activator of Transcription) pathway. In this focused review, we discussed studies related to the JAK-STAT pathway, but are cognizant that intricate interactions exist among them (Figure 1) [21]. Many of the aforementioned molecular pathways have been recently reviewed by others. Readers interested in those pathways are encouraged to read those excellent reviews [17,18,19,20].

The JAK-STAT pathway is a rapid membrane-to-nucleus signaling module [22] regulated by a wide array of extracellular signals including cytokines and growth factors, as well as cell-intrinsic mutations/alterations. Among all those upstream signals, interferons (IFNs), especially IFN-α/β (belonging to type I IFNs: IFN-Is) and IFN-γ (the only member in type II IFN), are the most widely studied [23]. With their pleiotropic immunological activities in almost all known pathophysiological settings, IFNs are the focal point of our review. Binding of IFN-α/β/γ to their respective receptors (IFN-α/β receptor 1/2 (IFNAR1/2) and IFN-γ receptor 1/2 (IFNGR1/2)) triggers receptor dimerization [24], which in turn recruits receptor-associated JAKs to close proximity (JAK1 and Tyk2 for IFN-α/β, and JAK1/2 for IFN-γ), culminating in JAK activation. Activated JAKs then phosphorylate tyrosine residues in the cytoplasmic tails of the receptors that serve as docking sites for STATs on the receptor–JAK protein complex, facilitating phosphorylation of a specific tyrosine residue in the C-terminus of STATs. Phosphorylated STATs subsequently dimerize with each other, through the SH2 (Src Homology 2)-domain, to form transcriptionally active complexes that translocate to the nucleus, bind to interferon-sensitive response elements (ISREs) or gamma-activated sites (GASs), and directly regulate the transcription of a wide array of interferon-stimulated genes (ISGs). There are seven members of the STAT family (STAT1, 2, 3, 4, 5a, 5b, and 6), and of them, STAT1/2 heterodimers are the most important mediators of the response to IFN-α/β by complexing with the IFN-regulatory factor 9 (IRF9) to form the ISG factor 3 (ISFG3) transcription factor complex; on the other hand, STAT1 homodimers are the major STAT proteins activated by IFN-γ [24]. Other STATs, namely STAT3 and STAT5, have also been found to be phosphorylated by IFN-α/β/γ [25,26,27]. While STAT4 and 6 can be involved, their activation is restricted to certain cell types such as endothelial cells [28,29]. For this reason, we will mainly consider STAT1, STAT3, and STAT5 proteins. In addition to these essential players in deciding the signaling strength and transcriptional regulation, the IFN–JAK–STAT axis is also delicately controlled by multiple negative regulators, including protein tyrosine phosphatases (PTPs) that block activation of JAK-STAT by dephosphorylating receptor complexes, suppressor of cytokine signaling (SOCS) proteins (e.g., SOCS1 and SCOS3) that directly interfere with the kinase domain of JAKs, and protein inhibitor of activated STAT proteins (PIAS) that suppress STAT transcriptional activity and regulate their degradation (Figure 1). Together, these construct an intricate and complicated network that controls the expression of various interferon-stimulated genes that are involved in cell proliferation and survival, cellular metabolism, DNA damage repair, and evasion from immunosurveillance [30,31,32,33]. As such, the IFN–JAK–STAT axis coordinates intercellular communication between tumor cells and stromal cells in the TME, controlling tumor progression and representing a central hub in governing tumor responses (sensitivity and resistance) to RT and IOs.

## 2. The IFN–JAK–STAT Axis in RT (Friends and Foes)

**IFN-α and β:** DNA is the primary target of RT. Upon RT, damaged DNA fragments that are released into the cytoplasm or exit into the extracellular space and, subsequently, engulfed by myeloid cells (e.g., DC) can bind to the cyclic GMP-AMP (cGAMP) synthase (cGAS). This, in turn, activates the adaptor protein STING (stimulator of interferon genes) and induces IFN-Is (especially IFN-β) in an interferon regulatory factor 3 (IRF3)/NK-κB-dependent manner [34,35]. Both DCs and tumor cells can produce IFN-Is after RT, although subsequent studies indicated that DCs were the major producers [36]. Consistent with this, IFNAR1 must be expressed by hematopoietic cells (including DCs) but not cancer cells to drive anti-tumor responses elicited by RT [37], establishing a pivotal role of the cGAS–STING–IFN-I–IFNAR1 axis in DCs in dictating therapeutic effects of RT. In addition to DNA damage, RT can also induce small endogenous non-coding RNAs (sncRNAs) such as small nuclear RNAs U1 and U2, which bind and activate the cytoplasmic viral RNA sensor RIG-I (retinoic acid-inducible gene I). Subsequently, the mitochondrial adaptor protein MAVS (mitochondrial antiviral-signaling protein) is activated and leads to enhanced IFN-I signaling, promoting radiosensitivity in murine embryonic fibroblasts (MEFs), glioblastoma (GBM), and colorectal cancer (CRC) [38]. Conversely, as a negative feedback mechanism, RT also upregulates the RNA helicase Laboratory of Genetics and Physiology 2 (LGP2), a suppressor of the RIG-I-MAVS pathway, which inhibits IFN-b expression and confers radioresistance to various tumor cell lines and MEFs [39]. Clearly, these on-and-off molecular switches of the IFN-I signaling after RT determine tumor responses to RT, highlighting a prominent role of the IFN-I signaling in RT efficacy. In support of this, IFN-Is directly enhance radiosensitivity in cervical cancer [40,41], non-small cell lung cancer [42], glioma [43,44], and pancreatic cancer [45]. Conversely, radioresistant glioblastoma (GBM) highly expresses negative regulators of the IFN–JAK–STAT signaling (SOCS1 and SOCS3) and attenuates IFN–JAK–STAT signaling [46]. Based on the radiosensitizing effects of IFN-I signaling and its importance in determining RT efficacy, clinical endeavors have been undertaken to test combination therapies of RT and IFN-Is. However, clinical outcomes from these trials were diverse and hard to explain. At present, it remains premature to conclude whether these combination therapies are effective or not, but early clinical trials with a limited number of melanoma patients reported severe subacute/late complications when treated with IFN-α+RT [47,48], casting doubts or requiring precautions when combining them in the clinic.

**IFN-γ:** RT also greatly increases the abundance of T cells in the tumor, through either promoting T cell infiltration [49] or survival/proliferation of pre-existing T cells [50]. A major effector molecule secreted by these tumor-infiltrating T cells (TILs) is IFN-γ. An early study showed that a single dose of RT (15 Gy) in B16-OVA melanoma significantly upregulated vascular cell adhesion molecule 1 (VCAM-1), monokine induced by gamma interferon (MIG, better known as CXCL9), CXCL10, and MHC-I; all are important immunostimulatory molecules in driving immune cell extravasation and infiltration. Interestingly, this was only observed in wild-type but not IFN-γ KO mice, providing direct evidence on the importance of host IFN-γ signaling in RT-driven immunological effects. In addition, deletion of IFNGR1 expression on melanoma cells also abrogated these RT effects, indicating an indispensable role of tumor IFNGR1 signaling in this process [51]. Using the same RT regimen in the MC38 CRC model, a subsequent study confirmed that host IFN-γ signaling was crucial for RT efficacy, but in contrast to melanoma, tumor IFNGR1 expression was largely dispensable, as transduction with dominant negative IFNGR1 did not affect RT efficacy [52]. Although this seemed to be contradictory to a far greater capacity of RT-treated TILs in lysing MC38 cells, in an IFN-γ-dependent fashion [52], one should consider that IFN-γ has pleiotropic functions and can reprogram TILs to become better effectors, which could, in turn, suppress the growth of MC38 cells via IFNGR1-independent mechanisms. On the other hand, IFN-γ directly induced upregulation of caspase-8 in human medulloblastoma cells, predisposing these cells to RT [53]. In human CRC cell lines, upregulation of IRF-1 (interferon regulatory factor-1), a transcriptional target of IFN-γ, inhibited cell proliferation and colony formation, enhancing the radiosensitivity of xenografts [54]. Moreover, typical fractionated RT (1.8–2 Gy/day, 5 days a week) activated the IFN-γ signaling pathway in A549 lung carcinoma through the MAPK pathway, leading to growth inhibition and cytotoxicity [55]. Taken together, the importance of tumor-intrinsic IFN-γ signaling in RT may be tumor type-dependent, although it is clear that host IFN-γ signaling plays an essential role in orchestrating the therapeutic effects of RT. In spite of this, substantial precaution needs to be given when combining IFN-γ with RT for therapeutic purpose, as an early phase IIIA/B clinical trial in patients with NSCLC reported severe toxicities including fatal radiation pneumonitis [56].

**STATs:** The above studies showed that IFN-α/β/γ, mainly produced by immune cells (e.g., DCs for IFN-α/β, and T cells for IFN-γ), have a largely “friend/helper” role in promoting RT efficacy. Based on this, one would speculate that downstream of IFN-α/β and IFN-γ, activation of STATs (particularly, STAT1) would exert a similar anti-tumor role. In support of this notion, anti-proliferative and pro-apoptotic effects of STAT1 in tumor cells were reported in early studies [57,58]. However, the RT effects were not investigated in those studies. Thus, a “friend/helper” role of STAT1 and other STAT proteins in RT remains to be established. To the contrary, most studies with RT so far have suggested a “foe” role of STATs, i.e., activated STATs in tumor are correlated with radioresistance. STAT3 has been shown to be a major mediator of resistance to RT and chemoradiotherapy in almost all common types of cancer [59]. One study reported that radioresistant human lung cancer cells had activated STAT3, coupled with increased expression of Bcl-xL, an important anti-apoptotic protein [60]. Another study using the radioresistant pancreatic cancer cells (PC) showed that STAT3 was also activated, but there was no overt upregulation of Bcl-xL; rather, the expression of survivin, an inhibitor of apoptosis was increased [61]. Despite these different mechanisms underpinning radioresistance in different types of cancer cells with active STAT3, targeting STAT3 was an effective approach to overcome radioresistance [60,62]. We previously found that inhibition of EGFR with cetuximab greatly improved the overall efficacy of RT in patients with head and neck squamous cell carcinoma (HNSCC) [63]. Considering that activation of STAT3 is an important downstream event of EGFR signaling, we tested whether targeting STAT3 could exert similar synergistic effects with RT. Our results indicated that either genetic knock-down of STAT3 or pharmacological inhibition of STAT3, especially when combined with a JAK inhibitor, greatly enhanced the radiosensitivity of HNSCC cells [64,65]. In addition to the important role of STAT3 activation in mediating radioresistance of HNSCC, another study found that in radioresistant HNSCC cells, STAT1 was the most highly expressed STAT protein and transduction of radiosensitive HNSCC cells with STAT1 induced radioresistance [66]. Conversely, STAT1 knock-down (STAT1^KD^) in both wild-type and radioresistant nu61 HNSCC led to tumor growth suppression and radiosensitization [67]. Mechanistically, STAT1^KD^ predisposed human HNSCC and melanoma cells to RT-driven suppression of glycolysis and lactate dehydrogenase A (a key enzyme in glycolysis), suggesting that STAT1 can protect tumor cells from RT-induced deprivation of energy and cell killing [67]. Interestingly, STAT1, but not STAT3 expression, was significantly increased in radioresistant human RCC samples and inhibition of STAT1 by fludarabine and siRNAs enhanced RCC cells’ radiosensitivity [68]. STAT5 also appears to impact the RT response. Inhibition of active STAT5 in prostate cancer led to downregulation of RAD51 (an essential homologous recombination (HR) DNA repair enzyme), which sensitized tumor but not normal tissue to RT [69]. Similarly, activation of STAT3 and STAT1/IRF1 by RT-induced double-strand DNA breaks (DSDBs) in different types of cancer can also interact with DNA damage repair enzymes such as Ataxia-Telangiectasia-Mutated (ATM)-Checkpoint kinase 1 (Chk1), ATR (ATM and Rad3-related)-Chk2, and BReast CAncer gene 1 (BRCA1). These interactions boosted DNA repair ability and upregulated PD-L1 (an important immunoregulatory molecule, discussed below), contributing to radioresistance [70]. Collectively, these studies indicate that STAT proteins act as “foes” to RT by promoting the metabolic fitness and DNA damage repair capability of tumor cells, which can be targeted to overcome radioresistance in cancer cells.

**Friends and Foes:** These seemingly contradictory friend and foe roles of IFNs and STATs in RT may reflect opposing effects of acute vs. chronic activation of the IFN–JAK–STAT axis associated with different RT regimens. It is possible that effective short-term RT regimens promote transient production of IFN-α/β by DCs and of IFN-γ by T cells, which temporarily engage STAT1 and/or other STAT molecules in both host cells (including immune cells) and tumor cells to trigger protective anti-tumor responses. In contrast, persistent or multiple rounds of RT caused persistent production of IFNs and overexpression of STAT1, associated with the formation and expansion of RT-resistant clones, as illustrated in HNSCC by Khodarev and colleagues [66,71]. In further support of this hypothesis, RT-induced expression level of type I IFN signaling genes was significantly higher in radioresistant breast cancer cells, as compared with nonresistant cells, which was dependent on the radiation dose and treatment time [72]. Additionally, IFN-related DNA damage resistance signature (IRDS) was associated with resistance to RT and that IRDS^+^ breast cancers were more likely to recur after mastectomy and adjuvant RT [73]. A similar finding was reported in IRDS^+^ GBM [74]. Furthermore, a recent study showed that continuous treatment of melanoma cells with IFN-γ, but not IFN-β, induced radioresistance, through both PD-L1-dependent and -independent mechanisms, and strategic targeting of the JAK-STAT with Ruxolitinib, a JAK1/2 inhibitor, could circumvent the radioresistance [75]. In addition to chronic/continuous exposure to IFNs, as aforementioned, constitutive activation of the JAK-STAT pathway could also occur as a result of stimulation of other extracellular signals (e.g., IL-6 and IL-10) [76], cell-intrinsic changes associated with non-receptor tyrosine kinases (NRTKs) such as Src and cytoplasmic kinases (protein kinase C) [77], as well as special cellular components in the TME (e.g., astrocytes in GBM) [78], contributing to radioresistance. Moreover, RT activated p53, which, in cooperation with oncogenic Ras in the unirradiated bystander cells, engaged the JAK-STAT axis and promoted tumorigenesis, conferring radioresistance to the unirradiated cells [79]. Intriguingly, although IFN-Is from DCs provided a dominant anti-tumor role after RT [36], IFN-Is in cancer cells actually protected them from CD8^+^ T cell-mediated killing after RT by inducing serpinb9, a physiological inhibitor of GzmB. This suggests that the cellular source of IFN-Is may dictate ensuing consequences [80]. Therefore, the duration of IFN stimulation, the source of IFNs, as well as other upstream signals/tumor cell-intrinsic alterations can converge on constitutive activation of the JAK-STAT pathway, resulting in radioresistance rather than radiosensitivity.

In summary, while there is a concern over the toxicity when IFN-α/β/γ is combined with RT in treating patients with cancer, inhibition of JAKs and STATs appears to be an attractive strategy to boost therapeutic efficacy of RT, considering their prominent roles in mediating radioresistance in various types of cancer. Nevertheless, only a limited number of clinical trials have explored the combination therapy of inhibition of JAK/STAT with RT and the results have been disappointing [81]. It is possible that this clinical information suffers from unselected patient enrollment to the trials, inappropriate scheduling of the JAK-STAT inhibitors and RT, doses and administration routes of the inhibitors, and specific RT regimens (dose and fractionation). Thus, although this combination therapy has a strong potential to generate synergistic outcomes, well-designed trials are needed to build optimal therapeutic plans.

## 3. The IFN–JAK–STAT Axis in IOs (The Yin and Yang)

**The Immunostimulatory Effects of the IFN–JAK–STAT Axis (The Yang)**: IFN-Is are crucial coordinators of innate and adaptive immunity, with well-recognized roles in viral immunity [82]. By either directly binding to IFNAR1/2 (Figure 1) or indirectly inducing production of other chemokines and cytokines, IFN-Is regulate immune cell trafficking, activation, and maturation [25], and are expected to be important for mediating the therapeutic effects of IOs. In fact, IFN-α2, one of the 13 types of human IFN-α, was the first cancer immunotherapy approved by the FDA in 1986. Since then, IFN-Is, almost exclusively IFN-α, have received additional approvals to treat several neoplastic diseases, including follicular non-Hodgkin lymphoma and resectable stage II and III melanoma, mainly in an adjuvant setting [83]. However, their use in cancer therapy has been dwindling in recent years, due to the limited efficacy and the revolutionary successes of other forms of IOs such as ICBs and ACT. To date, all seven FDA-approved ICBs target immune checkpoints CTLA-4 and PD-1/PD-L1 that are co-opted by tumor cells to evade immunosurveillance. By blocking these negative immune checkpoints, ICBs (anti-CTLA-4 and anti-PD-1/L1) re-invigorate the patients’ endogenous immune cells, particularly T cells to ward off tumor. ACT, on the other hand, utilizes exogenously generated super-T cells, including genetically manipulated autologous T cells, TILs, and chimeric antigen receptor-T cells (well known as CAR-T cells). Upon infusion into patients, these super-T cells can track down and eliminate tumor cells. Although the direct use of IFN-Is as immunotherapeutics in the clinic is decreasing, their important roles in mediating ICBs and ACT have been recognized. For example, activation of IFN-I signaling was shown to be correlated with abscopal effects of the combination therapy of anti-CTLA-4 and RT, which were abrogated when IFNAR1 was knocked down in tumor cells [84]. A separate study using a syngeneic lung cancer model reported that localized RT could overcome resistance to anti-PD-1 by inducing IFN-β production and subsequently enhancing MHC class I expression. This was largely abolished when mice were pretreated with anti-IFNAR1 blocking antibodies [85]. Further, it was found that TME of CRC induced a dramatic downregulation of IFNAR1 on CD8^+^ TILs and transferred CAR-T cells, leading to impaired survival of these cells and dampened anti-tumor responses of CAR-T and anti-PD-1 [86]. Interestingly, in-depth mechanistic studies revealed that IFNAR1 degradation was mediated by p38 MAPK, a kinase involved in ligand-independent downregulation of IFNAR1 [87]. Collectively, these studies establish a pivotal role of IFNAR1 expression (either on immune cells or tumor cells) in governing IO efficacy, but direct evidence of IFN-Is in this process is lacking, given that tumor expression of IFNAR1 can be regulated by ligand-independent mechanisms (e.g., MAPK) [87].

Like IFN-Is, IFN-γ also exerts a multitude of immunological activities, arguably with greater immunoregulatory potential than IFN-Is [88]. Pioneering work from the Schreiber group using either IFN-γ blocking antibodies [89] or mouse models lacking essential IFN-γ signaling molecules [90] provided compelling evidence on a pivotal role of the IFN-γ signaling in cancer immunosurveillance. Remarkably, a recent report found that the only shared feature from anti-CTLA-4, anti-PD-1, and the combination of both was the upregulation of *IFNG* gene expression, suggesting that IFN-γ might be the most important cytokine in ICBs [91]. To this end, we employed *IFNGR1*^−/−^ mice and adoptive transfer of total *IFNGR1*^−/−^ T cells to tumor-bearing mice, and found that loss of IFNGR1 in T cells completely abrogated the efficaciousness of combined anti-CTLA-4 and anti-PD-1 therapy [92]. Furthermore, we showed that melanoma patients that were not responsive to anti-CTLA-4 therapy had tumors harboring a greater copy number loss of IFN-γ signaling genes [93]. This was further confirmed in preclinical experiments using a B16 melanoma model with IFNGR1 knock-down (IFNGR1^KD^) that showed impaired IFN-γ signaling and reduced sensitivity to anti-CTLA-4 therapy [93]. In a separate study, Zaretsky et al. reported loss-of-function mutations of JAK1 and JAK2 in two out of four melanoma patients that developed therapeutic resistance to anti-PD-1; both JAK1/2 mutations were coupled with a total loss of functional response to IFN-γ and JAK1 mutation also caused loss of functional response to IFN-α/β [94]. These results were subsequently corroborated by a series of seminal studies in melanoma and CRC [95,96,97,98], pointing to an indispensable role of tumor-intrinsic IFN-γ signaling in ICBs. Despite these congruent findings, the specific underlying mechanisms of this ICB resistance are poorly defined, although it is generally agreed that IFN-γ induces MHC-I, MHC-II, and other important molecules in antigen-presenting machinery such as TAP (transporter associated with antigen processing) in tumor cells [99]. Intriguingly, a recent study showed that IFN-γ induced ER stress in various tumor cell lines, leading to impaired autophagic flux by post-transcriptional downregulation of LAMP-1/2 (lysosome membrane protein-1 and -2), key players in the maturation of autophagosome. This ER stress triggered unfolded protein responses (UPR), contributing to IFN-γ-induced apoptosis and cell cycle arrest [21]. In correlation, clinical data from patients with melanoma, HNSCC, and gastric cancer revealed that gene expression profiles (GEPs), including IFN-γ signature genes, successfully separated responders from non-responders to anti-PD-1 therapy [100]. More recently, another study showed that *Ifng* expression was correlated with longer progression-free survival of patients with melanoma and NSCLC treated with anti-PD-1 [101]. Besides these clinical correlations [100,101], there have been clinical efforts to engage the IFN–JAK–STAT pathway to boost therapeutic responses. A recent pilot clinical study reported that intratumoral administration of G100, a TLR4 agonist to patients with Merkel cell carcinoma can act through STAT1 and synergize with IFN-γ to induce promising therapeutic responses [102,103]. A number of ongoing clinical trials are examining a special formulation of TLR3 agonist capable of inducing IFNs and activating the JAK-STAT axis in various types of solid tumors, but the clinical outcomes remain to be reported [104]. Together, these results indicate that an intact IFN-γ signaling in both host T cells and tumor cells is required for therapeutic effects of IOs [105], with demonstrated clinical relevance and early promising clinical outcomes.

**The Immunosuppressive Effects of the IFN–JAK–STAT axis (The Yin):** In contrast to the positive roles of IFNAR1 expression and IFN-γ signaling in governing IOs, a recent study reported that prolonged IFN-γ (but not IFN-Is) exposure rendered B16-F10 melanoma cells resistant to ICBs+RT via STAT1-mediated epigenomic changes. Of note, although IFN-I was not required for the induction of resistance, it helped sustain ICB resistance, as additional IFNAR1 ablation on top of IFNGR1 deletion promoted a greater therapeutic response in ICB-resistant melanoma, as compared to IFNGR1 deletion alone. This study provided further support for the concept that prolonged IFN signaling induced ICB resistance, as administration of a JAK inhibitor to the delayed ICB therapeutic regimen (therefore, longer exposure to endogenous IFNs) restored the responsiveness to ICBs [75]. Notably, in this study, IFNGR1^KO^ B16-F10 cells did not show resistance to standard anti-CTLA-4 and anti-PD-L1, which, to some extent, contrasted with our study using IFNGR1^KD^ B16-BL6 cells [93]. This discrepancy may be due to the different melanoma models that were used, differential effects of IFNGR1^KO^ vs. IFNGR1^KD^ in tumor cells, and/or different types of ICBs (anti-CTLA-4+anti-PD-L1 vs. anti-CTLA-4 alone), which warrants further investigations. To this end, a recent study found that B16-F10 melanoma cells expressing the model antigen SIY (SIYRYYGL) with IFNGR1 KO or JAK1 KO were immunologically better controlled by immunocompetent hosts; however, when inoculated together with WT tumor cells, these KO cells escaped anti-PD-1 therapy and outgrew WT tumor cells, seemingly contributing to therapeutic resistance to ICBs [106]. Although it was shown that prolonged exposure to IFN-Is did not induce therapeutic resistance in B16-F10 melanoma to anti-CTLA-4 and RT [75], another study using different types of tumor models (sarcoma, CRC, and breast tumor) demonstrated that sustained IFN-I signaling induced secondary resistance to anti-PD-1 that was developed after four rounds of anti-PD-1 treatments. Mechanistically, this was due to IFN-I-induced upregulation of NOS2 (inducible nitric oxide synthase) but not PD-L1 upregulation [107]. Notwithstanding these discrepant findings of the role of IFN-I signaling in ICB resistance between these two studies, both indicated that deletion of IFNAR1 in tumor cells may not mediate primary resistance to ICBs. Namely, the first study showed a largely normal response in IFNAR1^KO^ B16-F10 melanoma [75] and the second study actually reported an improved response in IFNAR1^KO^ MCA205 sarcoma [107], which may be due to different tumor models or treatment regimens that were used in these studies. Nevertheless, these results indicate that prolonged IFN-I and IFN-γ signaling in tumors may dampen, rather than promote, therapeutic responses to ICBs.

In addition to the above-mentioned mechanisms of immunosuppression that are independent of PD-L1 upregulation [75,85,107], IFN-induced PD-L1 upregulation has been linked to immunosuppression in various tumor models. In this regard, IFN-β was shown to induce PD-L1 expression in LLC lung cancer cells in a JAK-dependent fashion [108]. Furthermore, in vitro treatment of gastric cancer cells with IFN-γ led to PD-L1 expression through JAK-STAT but not MAPK and PI3K pathway, couple with impaired cytotoxicity of tumor antigen-specific CD8^+^ T cells and EMT phenotype of tumor cells, which could be overcome by anti-PD-1 [109]. Similarly, in pancreatic ductal adenocarcinoma, IFN-γ induced PD-L1 expression and EMT, which was abrogated by siRNAs against STAT1, again highlighting an essential role of the JAK-STAT pathway [110]. Importantly, by blocking IFN-γ-driven PD-L1 upregulation via suppression of STAT3, two flavonoids (baicalein and baicalin) restored T cells’ capacity to kill hepatocellular carcinoma (HCC) cells in vitro and induced more significant tumor regression in immunocompetent BALB/c mice than in BALB/c-nu/nu mice that lack T cells, indicating an essential role of T cells in these therapies [111]. Clinically, PD-L1 upregulation via IFN-γ-JAK-STAT was more frequently observed in patients with larger CRC that were positive for vascular or lymphatic infiltration and poorly differentiated, in correlation to poor survival [112]. However, as we recently reviewed [105], despite numerous studies assessing PD-L1 upregulation as a predictive biomarker for therapeutic responses of ICBs, a definitive connection remains to be established. Based on this, we argue that PD-L1 expression on tumor cells is more of a dynamic and inducible biomarker, reflecting a greater likelihood of response to ICBs but not necessarily a consequence of surefire immunosuppressive mechanisms [105]. In keeping with this notion, a recent report demonstrated that PD-L1 expression in the presence of IFN-γ served as a biomarker for tumors’ sensitivity to IFN-γ treatment, but not immunosuppression [113]. That said, it remains possible that PD-L1 upregulation and accompanied signaling events after acute exposure to IFN correlate with ICB efficacy; conversely, chronic/sustained IFN signaling may induce epigenetic changes (e.g., STAT1-mediated epigenetic alterations [75]) that permanently fix these cells to be resistant to ICBs, regardless of their PD-L1 upregulation. However, despite the promising preclinical findings from targeting JAK-STAT [75] and supportive results in early patient trials, no JAK inhibitors have been approved by the FDA for cancer, due to their limited efficacy and high toxicity, and off-target immunosuppression [114,115]. Additional results are awaited from numerous ongoing clinical trials involving Ruxolitinib (a JAK1/2 inhibitor) and Tofacitinib (a JAK1/3 inhibitor), two FDA-approved JAK inhibitors for other clinical conditions [116].

There are other immunosuppressive mechanisms that can be engaged by IFNs. Using a lung adenocarcinoma model, Yu et al. [117] showed that IFN-γ production induced by anti-PD-1 led to nuclear phase separation and condensation of yes-associated protein (YAP), a downstream effector in the Hippo pathway. By complexing with TEA domain (TEAD) family members, YAP transcriptionally regulated thousands of genes (e.g., CD155), which negatively regulated T cell effector function and also recruited immunosuppressive cells such as tumor-associated macrophages (TAM), myeloid-derived suppressor cells (MDSC), FoxP3^+^ regulatory T cells (T_reg_), and cancer-associated fibroblasts (CAF) to establish an immunosuppressive TME. Importantly, the combination of YAP inhibitor with anti-PD-1 showed significant improvement on tumor suppression, independent of the STAT1-IRF1 axis and canonical ISGs, providing a new molecular circuit of IFN-γ-induced adaptive resistance to IOs that can be targeted to enhance ICB efficacy [117]. In contrast, induction of CD47 on tumor cells, a prominent “don’t-eat-me” signal, by IFN-γ relied on the JAK1–STAT1–IRF1 axis, which prevented phagocytosis of stressed tumor cells by macrophages. This finding was observed in various tumor cells, suggesting this is a universal immunosuppressive mechanism engaged by IFN-γ [118]. Furthermore, IFNs, in particular IFN-γ, can impact cellular metabolism by inducing tryptophan 2,3-dioxygenase (TDO) or indoleamine 2,3-dioxygenase (IDO), which catabolizes essential amino acid tryptophan to kynurenine. Kynurenine then activated the aryl hydrocarbon receptor (AhR) and induced T cell functional suppression and therapeutic resistance [119].

In summary, it has become clear that both immunostimulatory (Yang) and immunosuppressive (Yin) effects can be mediated by IFN signaling in response to IOs (Figure 2). However, the underlying molecular switches of these Yin–Yang outcomes are largely unknown, although it may be related to acute vs. chronic activation of the IFN–JAK–STAT axis, as discussed above in RT. In addition, a general belief is that STAT1 is closely associated with the immunogenic effect, and STAT3 and STAT5 behave more like oncogenic mediators [77], but no definitive evidence has been reported on this and it is likely that their specific roles are context dependent. Similarly, a recent study showed that JAK1 mediated IFN-γ-driven phosphorylation of STAT1/3/5 and induction of MHC-I, MHC-II, and PD-L1, whereas JAK2 was not involved in the activation of STAT1 and MHC-I/II induction but was essential for activation of STAT3/5 and PD-L1 upregulation. Importantly, inhibition of both JAK1/2 partially blocked anti-PD-L1 therapy, whereas selective blocking of JAK2 seemed to augment anti-PD-L1 therapy [120]. Furthermore, an early study showed that IFN-γ induced tumor dormancy when IFNGR1 expression level was low, but resulted in tumor elimination when it was high, indicating that the absolute expression level of IFNGR1 or signaling strength could determine the final outcomes of IFN-γ signaling [121]. Moreover, using a 3D fibrin gel model, a recent study found that IFN-γ induced apoptosis of differentiated tumor cells; however, if both IDO1 and AhR were highly expressed in the tumor-repopulating cells, IFN-γ induced p27 instead, which attenuated STAT1 signaling, preventing tumor cell death and promoting tumor cell dormancy [122]. Therefore, the final consequences of the IFN signaling may well depend on the duration (acute vs. chronic), signaling strength (the amount cytokine produced and/or the receptor expression level), the specific signaling molecules involved (JAK1-STAT1 vs. JAK2-STAT3/5), as well as tumor types and/or stages (early vs. late).

## 4. The IFN–JAK–STAT AXIS in RT and IOs

**The IFN–JAK–STAT Axis in Immunomodulatory Effects of RT:** As aforementioned, RT can induce DSDBs, sncRNAs, and DAMPs. While dsDNAs primarily activate the cGAS-STING pathway [123] and sncRNAs engage the RIG-I-MAVS pathway [38], different DAMPs are recognized by the host through different mechanisms. For example, translocated calreticulin (CRT) to the outer cell membrane (typically residing in the endoplasmic reticulum) promotes the phagocytosis of irradiated cells by macrophages and dendritic cells (DCs); HMGB1, a highly conserved nonhistone DNA-binding protein, acts as an agonist of Toll-like receptor 2 (TLR2) and TLR4 [124], two primary pattern recognition receptors (PRRs) on DCs; and extracellular ATPs bind to P2X7 purinergic receptors on DC [125,126]. All these danger/stress-associated signals converge on DC activation and maturation, leading to activation of tumor antigen-specific T cells. Two early studies showed that the IFNAR1 expression in DC (but not tumor cells) plays a dominant role in orchestrating the immunological effects of RT. However, it is noteworthy to point out that both studies employed a single high dose of radiation (20 Gy) [36,127], which may not be ideal to induce immunogenic effects. To this end, a recent study reported that single high-dose RT (20 or 30 Gy) actually induced expression of the three prime repair exonuclease 1 (Trex1) that degraded cytosolic dsDNAs and attenuated the cGAS-STING pathway in tumor cells, thereby lowering production of IFN-β by tumor cells. In contrast, hypofractionated RT (8 Gy/day on three consecutive days) did not induce Trex1 and, therefore, promoted stronger cGAS-STING activity as well as higher expression of IFN-β in tumor cells. Interestingly, knock-down of IFNAR1 in tumor cells completely abolished the therapeutic effects of this hypofractionated RT, indicating an indispensable role of tumor IFNAR1 expression in this system [84]. Thus, the relative importance of IFN-I signaling in tumor cells may be determined by RT dose and fractionation. Nevertheless, host cell expression of IFNAR1 has been consistently shown to be instrumental for RT-induced immunomodulatory effects [37,84,85,128], encompassing increased infiltration of CD45^+^ hematopoietic cells (CD4^+^ and CD8^+^ T cells, DCs, and macrophages, etc.), upregulation of CXCL10 [128] and CXCL16 [129] (chemokines for CXCR3^+^ and CXCR6^+^ effector T cells, respectively), increase in Fas ligand (FasL), production of effector molecules GzmB and IFN-γ [128], upregulation of MHC molecules [130], and downregulation of don’t-eat-me signal CD47 [131]. Similarly, host IFN-γ signaling has been shown to be essential in mediating anti-tumor immune responses elicited by RT using B16-OVA melanoma [51] and MC38 CRC [52]. While these studies compellingly pointed to a pivotal role of host IFN signaling in RT, it remains to be explored how the host JAK–STAT axis is involved in RT-induced immunological outcomes, although this is expected, given that JAKs and STATs have been shown to be crucial in dictating immune cell functions [22]. Further, a consensus on the role of tumor expression of IFNAR1 and IFNGR1 regarding RT response still cannot be reached, as either an essential role or a dispensable role was reported [36,84,127]. Additional work should be carried out to systematically assess how hyperfractionated RT, hypofractionated RT, and single-dose RT (high dose) may differentially impact the IFN-I and IFN-γ pathways in tumors, as distinct immunological outcomes may ensue after different RT regimens [36,84,127]. To this end, an early study reported that low-dose total-body irradiation (0.1–0.25 Gy, several times a week for a total dose of 1.5–2 Gy) induced impressive long-term tumor remission in the majority of patients with chronic lymphocytic leukemia and low-grade non-Hodgkin’s lymphoma. This was mediated by immune-enhancing mechanisms (e.g., T cell activation, increased production of IFN-γ and IL-2, and increased expression of IL-2R) rather than direct radiation killing [132]. Therefore, it may be possible that when RT dose is too high (>20 Gy), more immunosuppressive effects may be induced [84].

**The IFN–JAK–STAT Axis in Radiosensitizing Effects of IOs.** It has long been known that immunological competence of patients contributes to the efficacy of RT, which determines the dose of radiation required for effective tumor control [133]. T cells, particularly CD8^+^ T cells, are required for therapeutic effects of RT, as demonstrated by a study wherein deletion of CD8^+^ T cells largely abolished RT-induced growth suppression of primary melanoma [134]. A subsequent study pinpointed that it was the pre-existing intratumoral T cells but not newly infiltrated T cells that governed the therapeutic effects of RT [50]. Since IOs can rejuvenate the pre-existing T cells [92,135,136], it is expected that IOs could induce radiosensitizing effects and promote RT. In support of this idea, Rodriguez-Ruiz et al. showed that RT alone did not generate an overt abscopal effect (i.e., suppression of unirradiated tumors) in various tumor models (MC38 CRC, B16-OVA melanoma, and 4T1 triple-negative breast cancer), but adding ICBs to RT generated potent curative effects of both the irradiated and unirradiated tumors, achieving an impressive 100% cure of all the tumor-bearing mice [15]. Mechanistically, in vitro co-culture of EMT-6 breast cancer cells with activated CD8^+^ T cells greatly enhanced the radiosensitivity of EMT-6 cells, via IFN-γ production and iNOS upregulation [137]. Similarly, co-culture of Hela cells with CD4^+^ T cells radiosensitized Hela cells by promoting RT-induced G2/M arrest, in an IFN-γ-dependent fashion [138]. This IFN-γ-dependent mechanism was also confirmed in vivo. By promoting IFN-γ production by CD4^+^ and CD8^+^ T cells, ICBs significantly enhanced radiosensitivity [139,140]. IFN-γ, together with reduction of T_reg_ after ICBs, promoted recruitment of tumor-associated eosinophils (TAEs), which then mediated normalization of tumor blood vessels [141]. This alleviated severe hypoxia in the TME, arguably the most important mediator of radioresistance, and enhanced radiosensitivity. Convincingly, in vivo depletion of TAEs with anti-Siglec-F antibodies completely abrogated the therapeutic effect of anti-CTLA-4 [142]. Along this line, intratumoral injection of STING agonists also led to normalization of abnormal tumor vasculature, promoting infiltration of CD8^+^ T cells, in an IFN-I signaling-dependent fashion [143]. Interestingly, IFN-β itself was shown to possess anti-angiogenetic properties, and by upregulating angiopoietin 1 (Angpt1), suppressed abnormal angiogenesis and promoted tumor vascular maturation [144]. This, in turn, enhanced intratumoral oxygenation and radiosensitivity [145]. While these studies collectively showed that blood vessel normalization represented an important mechanism of IO-induced radiosensitization, we reason that IOs may drive the radiosensitizing effects by metabolically reprogramming the TME. A recent study elegantly showed that ICBs tilted the intratumoral metabolic tug-of-war between TILs and tumor cells towards T cells [146] and suppressed the Warburg effect, a well-known metabolic feature of tumor cells that preferentially utilize glycolysis for their bioenergetic and biosynthetic needs. The suppression of the Warburg effect in tumor cells reduced accumulation of lactate, pyruvate, and other antioxidants such as glutathione, thereby facilitating ROS production and oxidative stress induced by RT. Greater intracellular ROS level would help fix oxidative damage to DNA and enhance radiosensitivity. In support of this idea, knock-down of STAT1 predisposed nu61 HNSCC cells to RT-driven suppression of glycolysis and reduction in anti-oxidation capacity, leading to greater tumor growth suppression and radiosensitization [67], but it remains to be tested how IOs affect STAT1 and other members of the JAK-STAT family in tumor cells. In addition, IOs induced production of effector molecules in addition to IFNs such as TNF-α, perforin, and GzmB, and engaged other cell death pathways (e.g., FasL-Fas), which may facilitate or amplify the direct and indirect effects of RT, resulting in radiosensitization.

**The IFN–JAK–STAT Axis Underscoring Combination Therapy of RT and IOs:** With the immunogenic effects from RT and the radiosensitizing consequences from IOs, it is an appealing idea to combine RT and IOs as a strategy to improve therapeutic efficacy. To this end, adding anti-PD-L1 to RT significantly boosted overall efficacy and enhanced anti-tumor immunity, including increased production of IFN-γ in pancreatic ductal adenocarcinoma [147] and hepatocellular carcinoma [148]. Similar results were also found in TUBO mammary and MC38 CRC tumors when treated with anti-PD-L1 and RT, which greatly promoted CTL activation and production of IFN-γ and TNF-α. This, in turn, reduced infiltration of immunosuppressive MDSCs, primarily mediated by TNF-α and, to a minor extent, by IFN-γ [127]. Another study tested combination therapy of anti-CTLA-4 with RT and demonstrated that this strategy not only depleted T_reg_ (by anti-CTLA-4) but also promoted the diversity of TCR repertoire of TILs (by RT), resulting in potent anti-tumor responses. However, upregulation of PD-L1 on tumor cells rendered the majority of tumor-bearing mice resistant to RT+anti-CTLA-4, and additional blocking of PD-1/PD-L1 (triple-therapy) significantly improved the overall efficacy [16]. In a follow-up study, this same group reported a PD-L1-independent resistance mechanism that was regulated by STAT1-mediated epigenetic changes, which, when targeted with an inhibitor (Ruxolitinib), successfully overcame the therapeutic resistance [75]. Clinically, greater benefits have been reported in various tumor types when treated with combination therapies of RT+IOs, providing proof-of-concept clinical evidence [149]. From a mechanistic standpoint, although pre-existing intratumoral T cells play a predominant role in RT alone [50], both pre-existing and infiltrated T cells were required for the combination therapy of RT+IOs [150]. In keeping with an essential role of host IFNAR1 signaling in mediating RT and IOs, either use of IFNAR1^KO^ mice [84] or in vivo blocking of IFNAR1 with neutralizing antibodies [85] completely abolished the synergistic effects of RT+IOs. On the other hand, although T cells and host IFN-γ signaling are essential for RT [51] and ICBs [92], to the best of our knowledge, their importance in combination therapy of RT and IOs has not been established. Likewise, very limited efforts, if any, have been made to evaluate the role of the JAK-STAT in RT+IOs. Therefore, despite our improved mechanistic understanding of the role of IFN–JAK–STAT axis in RT+IOs, more work needs to be carried out to pinpoint their involvement in different experimental settings. To this end, another study reported that activation of STING after a single-dose 20 Gy RT in the MC38 CRC model actually drove an influx of immunosuppressive MDSCs and T_reg_, in an IFN-I and CCR2-dependent manner [151]. Moreover, RT, especially multiple rounds or prolonged regimens, can induce adaptive immunosuppression such as upregulation of PD-L1 and CD47 by promoting sustained IFN signaling in tumor cells [130] and production of immunosuppressive cytokine TGF-β [152] (the Yin effects of RT, Figure 3). Thus, rational strategies are needed to combine RT and IOs, by maximizing immunostimulatory effects and concomitantly minimizing immunosuppressive effects, which will greatly improve therapeutic efficacy. We argue the final outcomes may well depend on the specific tumor types and stages, radiation dose, fractionation, and types of RT (particles vs. photon), and the scheduling of RT and IOs.

## 5. Concluding Remarks

IOs, including ICBs and ACT, have emerged as a new pillar in cancer care, adding to well-established therapeutic modalities including RT. Whereas IOs have generated unprecedented long-term clinical benefits in some patients, their efficacy is still limited to a minority of cancer patients. RT, on the other hand, has been widely used to treat cancer (>50% patients), but patients have limited options when a recurrence arises. This has inspired great interest in combining IOs with RT to drive long-lasting therapeutic effects in more patients, which has prompted numerous endeavors to decipher the underlying mechanisms governing RT and IOs. One of such mechanisms is the IFN–JAK–STAT axis, which has been shown to play essential roles in mediating therapeutic sensitivity and resistance to RT and IOs. Several points are noteworthy. First, while essential roles of host IFNs signaling in dictating RT have been established [37,51,52,84,85,128], the actual role of tumor expression of IFNAR1 and IFNGR1 may depend on the dose/fraction of RT used [36,84,127] and the specific tumor types [51,52]. Further, constitutive activation of STATs, resulting from either prolonged IFN signaling, other extracellular stimulus (e.g., IL-10 and IL-6) [76], and tumor cell-intrinsic alterations, has been largely correlated with radioresistance. How to strategically promote immunostimulatory host IFNs and suppress immunosuppressive STATs will greatly improve RT efficacy. As of now, results from such clinical efforts have been largely disappointing [81]. We reason that guided enrollment of patients (e.g., selection of patients with the aberrantly active JAK-STAT pathway in tumor cells), rational scheduling of JAK-STAT inhibitors with RT [65], as well as RT dose and fractionation (single high-dose RT may not be ideal [84,151]). Second, a pivotal role of IFNAR1 and IFNGR1 expression (either on immune cells or tumor cells) in governing IOs has been established, but prolonged/sustained IFN signaling could mediate resistance to IOs, either through negative feedback mechanisms such as upregulation of PD-L1, CD47, and YAP, or through epigenetic modifications in the tumor cells. In addition, the net outcomes from the IFN signaling may also depend on the signaling strength (lFNGR1^High^ vs. IFNGR1^Low^), the specific signaling molecules involved (JAK1-STAT1 vs. JAK2-STAT3/5), and tumor types and/or stages (more advanced large tumors with highly activated JAK-STAT tend to develop worse responses). Third, additive/synergistic clinical outcomes have been reported from clinical trials with combination therapy of RT and IOs. Nevertheless, the overall efficacy from RT+IO therapies needs to be improved. Clearly, the IFN–JAK–STAT axis, especially the IFN-γ signaling pathway, plays a very important role in mediating immunogenic effects of RT and in orchestrating radiosensitizing effects of IOs, placing it at the central stage of connecting RT and IOs. Thus, therapeutic targeting of this pathway with combinatorial therapies of RT and IOs would lend great value in future design of clinical trials. In this regard, several prominent factors worth considering are sequential administration of RT and IOs, RT doses and fractionation (hyperfractionation or hypofractionation), the specific types of IR (proton/heavy particles vs. photon), RT therapeutic plans (inclusion or exclusion of nodes), and dose rates (super-high dose rate or conventional dose rate). Additionally, the inclusion of more than one ICB with RT may be considered, as some preclinical studies showed that concurrent administration of anti-PD-L1 [153] and pre-treatment with anti-CTLA-4 [154] induced the best clinical responses, when combined with RT. Importantly, retrospective analysis of clinical trials with RT and anti-CTLA-4 also indicated that initiation of anti-CTLA-4 prior to RT greatly improved overall survival [155]. Furthermore, recent innovations with ultra-high dose rate RT (FLASH) have shown promise in sparing normal tissue [156] while killing tumors [157], which should be explored in combination with IOs. Additional mechanistic understanding of how the members of the IFN–JAK–STAT functionally and physically interact with each other to mediate the immunological effects of RT/FLASH-RT and radiosensitizing effects of IO will be instrumental in conceiving optimal therapeutic plans. In this regard, a reciprocal regulation between STAT3 and STAT1 in tumor cells has been discussed [77] and that different upstream signals (IFN-β vs. oncostatin-M [158]) or even the IFN-βs from different cells [80] may have opposite effects in response to RT and IOs. All in all, these studies imply a Goldilocks principle regarding the role of the IFN–JAK–STAT axis in RT and IOs, that is, too little (lack of/impairment) [37,51,52,84,85,93,94,128] or too much (sustained/chronic activation) [75,107] may dampen the optimal therapeutic effects of RT and IOs.

## Figures and Tables

**Figure 1 ijms-22-12295-f001:**
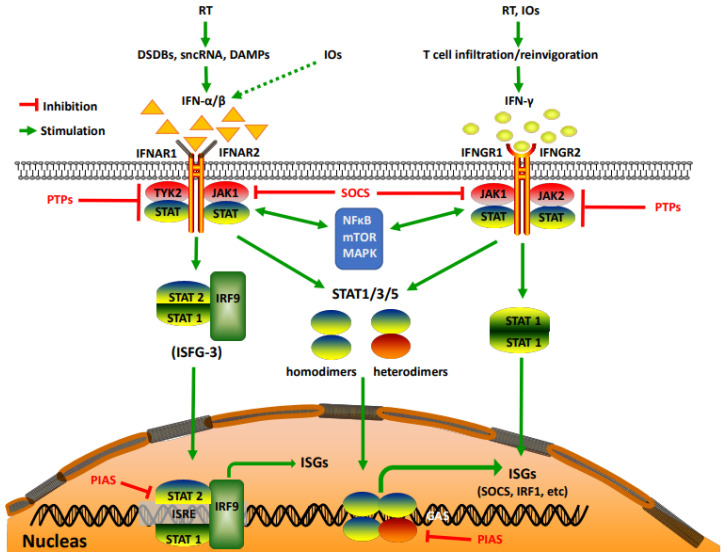
Activation of the IFN–JAK–STAT pathway by RT and IOs. DSDBs: double-strand DNA breaks; sncRNAs: small endogenous non-coding RNAs; DAMPs: damage-associated molecular patterns; IRF1 and IRF9: IFN-regulatory factor 1 and 9; ISGs: interferon-stimulated genes; ISFG3: ISG factor 3; PTPs: protein tyrosine phosphatases; SOCS: suppressor of cytokine signaling; PIAS: protein inhibitor of activated STAT proteins.

**Figure 2 ijms-22-12295-f002:**
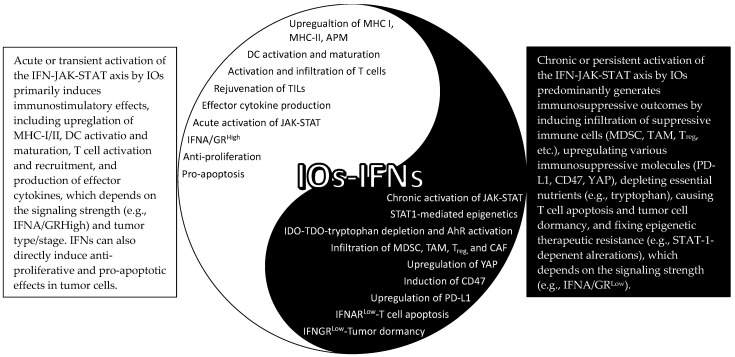
The Yin–Yang effects of the IFN–JAK–STAT in IOs. APM: antigen processing machinery; TILs: tumor-infiltrating T cells; YAP: yes-associated protein; MDSC: myeloid-derived suppressor cells; TAM: tumor-associated macrophages; CAF: cancer-associated fibroblasts; IDO/TDO: indoleamine/tryptophan 2,3-dioxygenase; AhR: aryl hydrocarbon receptor.

**Figure 3 ijms-22-12295-f003:**
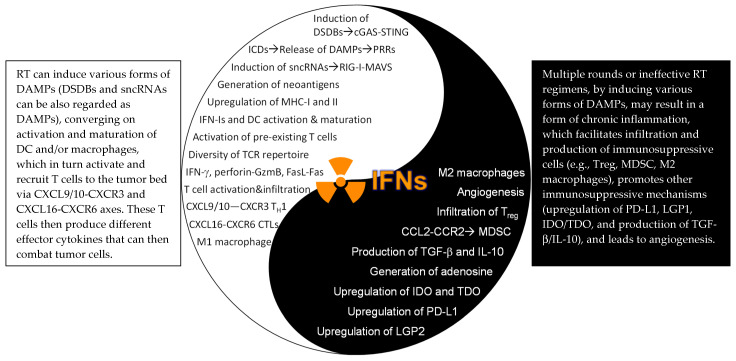
The Yin–Yang effects of the IFN–JAK–STAT in RT. DSDBs: double-strand DNA breaks; cGAS: cyclic GMP-AMP (cGAMP) synthase; STING: stimulator of interferon genes; ICDs: immunogenic cell death: DAMPs: damage-associated molecular patterns: sncRNAs: small endogenous non-coding RNAs; TH1: IFN-γ-producing type I CD4+ T cells; CTLs: cytotoxic T lymphocytes (CD8+ T cells); LGP2: laboratory of genetics and physiology 2; IDO/TDO: indoleamine/tryptophan 2,3-dioxygenase; MDSC: myeloid-derived suppressor cells.

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
