# Peer review of "Bridging Radiotherapy to Immunotherapy: The IFN–JAK–STAT Axis"

_ijms, 2021, doi:10.3390/ijms222212295_

Round 1
Reviewer 1 Report
Overall the study does not have any problem. It is a narrative review focusing on the immunomodulatory pathway modifications following radiotherapy. The main problem with the paper is the fact that is a narrative review focusing on this concept of immune activation/immunosuppression both generated by radiotherapy using this Yin/Yang concept....still, it is interesting and most of the information is useful, although nothing really new is reported. Some revisions may be necessary before this article may be considered for publication;
All the introduction basically lacks bibliographical reference; I suggest the authors add some references.
Although it is a narrative review, a materials and methods section explaining how you selected studies and what keywords you used would be a great addition.
Figures 2 and 3 are a little bit confusing...I think it would be better if sentences would be outside of the circles.
Thank You
Author Response
We thank this reviewer for the insightful evaluation of our review. As this reviewer pointed out, this is a rather complicated field. We proposed some ideas (e.g., acute vs chronic activation of the IFN-JAK-STAT pathway, RT dose and fractionation, etc.) to provide reasonable explanations for the Yin and Yang immunomodulating effects, which await further investigations. We have addressed all the comments and our responses (or excerpts from our revised manuscripts) are listed below.
Comment: All the introduction basically lacks bibliographical reference; I suggest the authors add some references.
Response: We have included more relevant references (in total, 18 more).
Comment: Although it is a narrative review, a materials and methods section explaining how you selected studies and what keywords you used would be a great addition.
Response: We described how we conducted the literature search in the Abstract and then in the Introduction. "We therefore conducted a systemic literature search on the molecular underpinnings of RT-induced immunomodulation and IOs, which identified the IFN-JAK-STAT pathway as a major regulator. Our further analysis of the relevant studies revealed that the signaling strength and duration of this pathway in response to RT and IOs may determine the eventual immunological outcomes."
Comment: Figures 2 and 3 are a little bit confusing...I think it would be better if sentences would be outside of the circles.
Response: We have added two text boxes to Figures 2 and 3. Please refer to the new figures in the revision.
Reviewer 2 Report
The manuscript entitlred:" BRIDGING RADIOTHERAPY TO IMMUNOTHERAPY: THE 2 IFN-JAK-STAT AXIS" focused on a systemic revision of literature data about the role of biomarkers in the prediction of clinical outcome in tumor patuients under radio/chemotherapy approach is well written and requires some minor integrations to be accpeted for the publication
- Please, could the authors report other experiences where these biomarkers play a crucial role in the clinical practice of soldi tumor patients?
- Could the authors also report clinical trail where these biomarkers are investigated in the administration of solid tumor patients?
- Could the authors also report other clinical details according to parapgraph entitled;"The IFN-JAK-STAT Axis in Radiosensitizing Effects of IOs"? In details i would suggest to describe clincial data in order to improve the qualityu of the manuscript
Author Response
We thank this review for the constructive comments, which we have addressed accordingly, as follows,
Comment: Please, could the authors report other experiences where these biomarkers play a crucial role in the clinical practice of soldi tumor patients?
Response (Lines 344-351 in our revised manuscript): Besides these clinical correlations[100, 101], there have been clinical efforts to engage the IFN-JAK-STAT pathway to boost therapeutic responses. A recent pilot clinical study reported that intratumoral administration of G100, a TLR4 agonist to patients with Merkel cell carcinoma can act through STAT1 and synergize with IFN-g to induce promising therapeutic responses[102, 103]. A number of ongoing clinical trials are examining a special formulation of TLR3 agonist capable of inducing IFNs and activating the JAK-STAT axis in various types of solid tumors, but the clinical outcomes remain to be reported[104].
Comment: Could the authors also report clinical trail where these biomarkers are investigated in the administration of solid tumor patients?
Response (Lines 416-421 in our revised manuscript): However, despite the promising preclinical findings from targeting JAK-STAT[75] and supportive results in early patient trials, no JAK inhibitors have been approved by the FDA for cancer, due to their limited efficacy and high toxicity, and off-target immunosuppression[114, 115]. Additional results are awaiting from numerous ongoing clinical trials involving Ruxolitinib (a JAK1/2 inhibitor) and Tofacitinib ( a JAK1/3 inhibitor), two FDA approved JAK inhibitors for other clinical conditions[116].
Comment: Could the authors also report other clinical details according to parapgraph entitled;"The IFN-JAK-STAT Axis in Radiosensitizing Effects of IOs"? In details i would suggest to describe clincial data in order to improve the qualityu of the manuscript
Response: In our original submission, we referred "radiosensitizing effects of IOs" to the greatly improved therapeutic efficacy when IOs were combined with RT, as compared to RT alone. We cited a review article that specifically discussed about this "Wang, Y.; Deng, W.; Li, N.; Neri, S.; Sharma, A.; Jiang, W.; Lin, S. H., Combining Immunotherapy and Radiotherapy for Cancer Treatment: Current Challenges and Future Directions. Front Pharmacol 2018, 9, 185." Our discussions in the section of "The IFN-JAK-STAT Axis in Radiosensitizing Effects of IOs" were on the potential underlying mechanisms from the preclinical studies, which haven't been confirmed/correlated in the clinical setting. Hope this helps.